# Cardio-Pulmonary Dysfunction Evaluation in Patients with Persistent Post-COVID-19 Headache

**DOI:** 10.3390/ijerph19073961

**Published:** 2022-03-26

**Authors:** Álvaro Aparisi, Cristina Ybarra-Falcón, Carolina Iglesias-Echeverría, Mario García-Gómez, Marta Marcos-Mangas, Gonzalo Valle-Peñacoba, Manuel Carrasco-Moraleja, César Fernández-de-las-Peñas, Ángel L. Guerrero, David García-Azorín

**Affiliations:** 1Department of Cardiology, Hospital del Mar, 08003 Barcelona, Spain; alvaro_aparisi@hotmail.com; 2Department of Cardiology, Hospital Clínico Universitario de Valladolid, 47005 Valladolid, Spain; mcdeybarrafalcon@gmail.com (C.Y.-F.); caroliglesiasecheverria@gmail.com (C.I.-E.); mariogg8@outlook.com (M.G.-G.); martamarcosma@outlook.com (M.M.-M.); manuelhcuv@gmail.com (M.C.-M.); 3Department of Neurology, Hospital Clínico Universitario de Valladolid, 47005 Valladolid, Spain; gvpenacoba@gmail.com (G.V.-P.); gueneurol@gmail.com (Á.L.G.); davilink@hotmail.com (D.G.-A.); 4Department of Physical Therapy, Occupational Therapy, Physical Medicine and Rehabilitation, Universidad Rey Juan Carlos (URJC), 28922 Madrid, Spain

**Keywords:** long-COVID syndrome, cardio-pulmonary exercise testing, headache disorders, COVID-19

## Abstract

Background (1): Headache is a prevalent symptom experienced during ongoing SARS-CoV-2 infection, but also weeks after recovery. Whether cardio-pulmonary dysfunction contributes causally to headache persistence is unknown. Methods (2): We conducted a case-control analysis nested in a prospective cohort study. Individuals were recruited from August 2020 to December 2020. Patients were grouped according to the presence or absence of long-COVID headache for three months after COVID-19 resolution. We compared demographic data, clinical variables, cardio-pulmonary laboratory biomarkers, quality of life, and cardio-pulmonary function between groups. Results (3): A cohort of 70 COVID-19 patients was evaluated. Patients with headaches (n = 10; 14.3%) were more frequently female (100% vs. 58.4%; *p* = 0.011) and younger (46.9 ± 8.45 vs. 56.13 ± 12 years; *p* = 0.023). No between-group differences in laboratory analysis, resting echocardiography, cardio-pulmonary exercise test, or pulmonary function tests were observed. Conclusion (4): In this exploratory study, no significant differences in cardio-pulmonary dysfunction were observed between patients with and without long-COVID headache during mid-term follow-up.

## 1. Introduction

Headache is a frequent symptom in acute systemic infections, including acute respiratory tract infections such as influenza or Epstein–Barr virus infection [1,2]. In the acute phase of coronavirus disease 2019 (COVID-19), between 25 and 72% of patients suffer from headaches [3,4]. Once the acute infection is resolved, headache is also a common post-COVID-19 symptom [5]. The proportion of patients with headaches during the acute phase that report persistence over time is estimated to be around 25% after three months and 16% after nine months [6,7,8]. It has also been observed that the headaches tend to have a migraine phenotype [9].

COVID-19 is a systemic disease with a complex pathophysiological mechanism that includes endothelial damage, microvascular injury, and thrombosis, among other effects [10]. Recently, a new entity known as long COVID has emerged, defined as the persistence of symptoms beyond 4 weeks for the subacute or more than 12 weeks for the chronic presentations [11]. The WHO recently proposed a clinical-case definition for the post-COVID-19 condition, setting 3 months as the usual time from the onset of COVID-19, acknowledging that symptoms may fluctuate or relapse over time [12]. A recent study observed that when headache persists beyond 2 months after the acute phase of the disease, it adopts a chronic pattern [8]. We recently reported an impaired ventilatory efficiency among patients with long-COVID dyspnea, with 14.3% of the global population presenting long-lasting headache [13].

The pathophysiology of long-COVID headache is not completely known. During the acute phase of the disease, it has been associated with a better prognosis and lower mortality [3,4,5,14]; however, better quality data are needed [15]. Patients with headaches have shown a different clinical presentation and variable levels of biomarkers, including C-reactive protein, D-dimer, or interleukine-6, all of which are associated with COVID-19 severity [3,14,16]. The cytokine profile of patients suffering from headache as an onset symptom was also different from that of patients with COVID-19 without headache [17].

Current evidence suggests that hypercapnia, hypoxemia, and cardiac failure could have a pathogenic role in some forms of headache [18,19,20,21]. Ventilatory inefficiency is associated with cardio-pulmonary conditions and gas-exchange anomalies. To address this hypothesis, we performed an exploratory cohort study to test whether COVID-19 survivors with long-COVID headache had cardio-pulmonary dysfunction more frequently than COVID-19 patients without headache. Therefore, this study aimed to evaluate whether patients with long-COVID headache, evaluated three months after infection, exhibited a higher frequency of abnormalities in cardio-pulmonary function than those patients without headache.

## 2. Materials and Methods

### 2.1. Study Design and Patient Selection

The present study is a post hoc analysis of a prospective cohort study, conducted in a third-level university public hospital (University Hospital of Valladolid) with a reference population of 260,000 patients. The study protocol was registered in ClinicalTrials.gov (NCT04689490), and the main purpose was to evaluate the causes and consequences of persistent dyspnea in COVID-19 survivors. The study period included patients managed in the hospital and the outpatient clinic from August 2020 to December 2020. Patient recruitment was non-probabilistic. All consecutive COVID-19 survivors that were evaluated in a specific COVID-19 consultation in an appointment three months after COVID-19 resolution were invited to participate.

### 2.2. Eligibility Criteria

Patients were included if: (1) they had a confirmed respiratory syndrome coronavirus-2 (SARS-CoV-2) infection, according to World Health Organization protocols, identified through positive oropharyngeal RT-PCR [22]; (2) had survived three months after the event; and (3) agreed to participate and signed an informed consent form. Exclusion criteria included: (1) age < 18 years old; (2) pregnancy or breastfeeding; (3) terminally ill patients with a reduced life expectancy during the follow-up period; (4) active SARS-CoV-2 infection; (5) inability to exercise due to post-COVID-19 sequelae; or (6) prior history of severe cardio-pulmonary disease.

Patients were classified as cases or controls depending on the presence or absence of long-COVID headache. The presence of headache was defined as any new-onset post-COVID-19 headache, or the worsening of a prior headache disorder before the infection, defined as a two-fold or greater increase in frequency and/or severity, according to the International Classification of Headache Disorders, 3rd version (ICHD-3) [23].

### 2.3. Procedure

The eligible patients underwent a standardized evaluation within 30 days after the screening visit. All participants had a mandatory RT-PCR for SARS-CoV-2 48 h before evaluation. The study consisted of an in-person clinical interview assessing demographic and clinical variables, a quality-of-life evaluation, and a comprehensive cardio-pulmonary function assessment including an echocardiogram, a pulmonary function test by spirometry, and cardio-pulmonary exercise tests (CPET). A series of cardio-pulmonary laboratory biomarkers and low-grade inflammation biomarkers were determined. Figure 1 summarizes the study procedures. The total duration of the evaluation was approximately 120 min per patient.

#### 2.3.1. Demographics and Clinical Data

The demographic variables captured were age, sex, body mass index, and body surface area. An accurate history of previous medical comorbidities was taken, e.g., hypertension, dyslipidemia, chronic kidney disease, or prior history of cardio-pulmonary or cerebrovascular diseases. We assessed whether participants were hospitalized or managed in an outpatient setting. Details of the received treatments, whenever applied, were collected, including anticoagulation, glucocorticoids, statins, hydroxychloroquine, azithromycin, and lopinavir/ritonavir. The presence of post-COVID-19 symptoms during the follow-up, including anosmia, chest pain, dysgeusia, dyspnea, fatigue, myalgia, neurological symptoms, sensory disturbances, or palpitations was recorded using a standardized questionnaire. Patients with long-COVID headache were followed in neurological outpatient clinics until resolution.

#### 2.3.2. Quality-of-Life Questionnaire

A comprehensive integrated assessment of symptom burden was carried out using the Kansas City Cardiomyopathy Questionnaire (KCCQ) [24]. This questionnaire was developed to quantify the quality of life (QoL) of heart failure patients and, in the current study, was used for assessing the impact of cardio-pulmonary symptoms in post-COVID-19 patients. The KCCQ has 23 items that are summarized into a single global score encompassing 7 items: symptom frequency, symptom burden, symptom stability, physical limitations, social limitations, quality of life, and self-efficacy. Scores range from 0 to 100, where lower scores are associated with severe limitations and vice versa [24].

#### 2.3.3. Pulmonary Function Test

Pulmonary function was assessed by spirometry, and the following parameters were analyzed: forced expiratory volume in the first second (FEV1%), forced vital capacity (FVC%), FEV1/FVC, total lung capacity (TLC%), and diffusing capacity (DLCO% and KCO%), according to the recommendations of the European Respiratory Society [25,26].

#### 2.3.4. Resting Echocardiography

Echocardiographic variables were collected and analyzed offline (EchoPAC software, version 202) by two independent observers. Images were recorded on each of the standard projections in compliance with the recommendations of the American and European Societies of Echocardiography [27]. Cardiac function (systolic, diastolic, and longitudinal strain) and dimensions were quantified. Other parameters assessed were the concomitant presence of valvular disease and regional wall motion abnormalities.

#### 2.3.5. Cardio-Pulmonary Exercise Test

All CPETs were supervised by an experienced physician and performed using a progressive incremental ramp protocol on a treadmill (Marquette Max-1 treadmill, Marquette Electronics Inc., Milwaukee, WI, USA) integrated with a metabolic system (CPX Express, Medgraphics, Cardiorespiratory Diagnostic Systems, Medical Graphics Corporation, St Paul, MN, USA) until patients reached physical exhaustion or until they reached their maximal capacity. There was continuous monitoring of cardiac parameters and peripheral oxygen saturation during the test. Ventilation efficiency and aerobic capacity were evaluated during anaerobic threshold and peak exercise capacity. The test was stopped if sudden arrhythmias, hypotension (defined as a systolic blood pressure decrease >10 mmHg), repolarization abnormalities, or symptoms suggestive of myocardial ischemia appeared [28].

#### 2.3.6. Laboratory Biomarkers

A fasting blood test was collected from all included patients from the right antecubital vein. The evaluated parameters and biomarkers included lymphocytes (reference value (RV): cell count × 10^9^/L), hemoglobin (RV: 12–16 g/dL), platelets (RV: 150–400 count × 10^9^/L), lactate dehydrogenase (LDH) level (135–250 U/L), serum creatinine (RV: 0.5–1.1 mg/dL), aspartate aminotransferase (AST) (RV < 32 U/L), (RV < 40 U L); D-dimer (RV < 500 ng/dL), serum ferritin (RV: 15–150 ng/mL), interleukin-6 (RV: <5.9 pg/mL), high-sensitivity C-reactive protein (hs-CRP) (RV 1–5 mg/L), NT-ProBNP (0–125 pg/mL), and high-sensitivity Troponin T (hs-TnT) (RV: <14 pg/mL).

All tests were performed in a certified clinical laboratory (ISO 9001:2015). Inflammatory markers were evaluated using a particle-enhanced immunoturbidimetric, colorimetric (e501 Module Analyzer^®^, Roche Diagnostics, Basel, Switzerland) immunoassay system on IMMULITE^®^ 2000 for interleukin-6 (IMMULITE^®^ 2000 IL-6, Siemens Healthcare Diagnostic, Marburg, Germany). Cardiac biomarkers were measured by electrochemiluminescence immunoassay with a cobas^®^ 6000 c 601 analyzer (Roche Diagnostics, Basel, Switzerland). D-dimer results were obtained by a turbidimetric test using an ACL TOP 500^®^ hemostasis testing system (Werfen Company, Cuenca, Spain).

### 2.4. Ethics

The local ethics committee of the East Valladolid Health Care area approved the study protocol (CASVE PI-20-1894), and all participants provided written informed consent. The study followed the guidelines of the Declaration of Helsinki of the World Medical Association. Existing datasets were recorded in an electronic database, and all personal details were anonymized.

### 2.5. Statistical Analysis

Categorical variables are reported as absolute values and percentages, whereas continuous variables are expressed as medians (interquartile range (IQR)) or standard deviations (SD)). The normality of continuous variables was verified with the Kolmogorov–Smirnov test and Q–Q plots. Categorical variables were compared using the chi-square test and the Fisher exact test when necessary. We compared continuous variables with Student’s *t*-test or the Mann–Whitney U test. The statistical analyses were performed using SPSS Statistics, version 26.0 (Armonk, NY: IBM Corp.). Differences were statistically significant when the *p*-value was < 0.05. Missing data were managed using complete-case analysis. The sample size was not estimated in advance and the analysis was performed with all the available data.

## 3. Results

### 3.1. Participants

During the study period, 70 patients fulfilled the eligibility criteria. All the enrolled patients completed the study. Ten (14.3%) patients had long-COVID headache, with a mean total duration of 9 (SD: 5) months until complete resolution.

The demographic and clinical characteristics are summarized in Table 1. Patients with headaches were more frequently female (100% vs. 58.4%; *p* = 0.011) and younger (46.9 ± 8.45 vs. 56.13 ± 12 years; *p* = 0.023). There were no statistically significant differences in the presence of previous comorbidities. Additionally, patients with headaches also experienced long-COVID myalgia (30% vs. 5%; *p* = 0.034) and olfactory abnormalities (30% vs. 5%; *p* = 0.034) more frequently than those without headaches.

### 3.2. Laboratory Biomarkers, Echocardiographic Findings, and Quality of Life

Laboratory, echocardiographic, and QoL data are listed in Table 2. Patients with headache had a lower median value of hs-TnT (3 (3–3.7) vs. 5.6 (3.9–7.7); *p* = 0.029) and higher levels of fibrinogen (498 (468–504) vs. 421 (367–466); *p* = 0.028). There was a trend towards significance regarding ferritin levels (55.6 (30.5–123) vs. 122 (55.4–171.3), *p* = 0.052). No other significant differences regarding other biomarkers or laboratory parameters were observed.

All patients had a preserved systolic and diastolic function, with no significant between-group differences, with the exception of the E/A ratio (1.27 (1.06–1.48) vs. 0.88 (0.5–0.8); *p* = 0.01), which was also within normal values. We did not observe significant valvulopathies, regional wall abnormalities, or other resting echocardiographic effects suggestive of cardiac sequelae.

Concerning the quality of life, no significant differences were observed in the KCCQ except in the social domain (49.5 ± 25.1 vs. 68.4 ± 21.2; *p* = 0.014).

### 3.3. Pulmonary Function Test and Cardio-Pulmonary Exercise Test

The findings of the cardio-pulmonary evaluation are shown in Table 3. There were no significant differences in the CPET main performance variables between groups, but lower systolic blood pressure was observed before (122 (109–140) vs. 140 (126–147) mmHg; *p* < 0.001) and after (139 (134–143) vs. 163 (151–180) mmHg; *p* < 0.001) CPET in patients with post-COVID-19 headache. There was also a trend towards a lower peak diastolic blood pressure (80 (77–90) vs. 91 (85–100), *p* = 0.059) in long-COVID headache patients. Respiratory variables were comparable between groups. Neither desaturation nor headache occurred during effort in any of the patients.

## 4. Discussion

In this exploratory study, we explored the hypothesis of potential cardio-pulmonary dysfunction as a cause of long-COVID headache. This is the first study that has specifically assessed this topic. The main findings of this study may be summarized as follows: (1) approximately 14.3% of the patients studied had long-COVID headache; and (2) we did not observe any overall statistically significant differences regarding the presence of abnormal cardio-pulmonary biomarkers or functional parameters evaluated by different modalities between COVID-19 survivors. Thus, cardio-pulmonary dysfunction may not be a key factor in long-COVID headache pathophysiology.

Patients with long-COVID headache were younger and more frequently female, had a higher frequency of anosmia and myalgia, and presented higher values of fibrinogen and lower ferritin levels. These results are in line with results from other series of patients with long-COVID headache and may be related to the fact that headache is considered to be a symptom associated with a more efficient immune response [3,4,5,14].

The quality of life of COVID-19 survivors may be influenced by the persistence of post-COVID-19 symptoms. The Kansas City Cardiomyopathy Questionnaire evaluates the impact of cardio-pulmonary dysfunction symptoms on different domains [24]. We observed a similar global score, but with a significant limitation on the social domain among those with long-COVID headache. However, we did not find the greater rate of dyspnea or fatigue that could explain such a perception. Therefore, the results for quality of life may highlight the disability caused by headache in post-COVID-19 patients. In the case of headache sufferers, the pandemic itself was associated with higher levels of stress, affective symptoms, and insomnia, which could impact on the headache prognosis and course over time [29].

Cardiac dysfunction has been reported as a possible cause of headaches [30,31]. Cardiac cephalalgia is defined in the ICHD-3 as headache developed in temporal relation to the onset of acute myocardial ischemia that significantly worsens or improves in parallel with the worsening/improvement of the myocardial ischemia [23]. Acute myocardial injury has been described during the acute phase of COVID-19 in up to 7–40% of patients [32,33]. Potential causes include direct viral invasion of the myocardium to oxygen supply–demand imbalances [32,33]. To date, no evidence of chronic myocardial injury has yet been found yet in COVID-19 survivors with persistent symptoms after the acute phase of the disease. In our sample, we did not observe any direct or indirect sign of ischemia in cardiac biomarkers, resting echocardiography, or the cardio-pulmonary exercise test. The only difference observed in the echocardiogram was the E/A ratio, probably explained by the between-groups age difference. Furthermore, none of the patients developed headache during the CPET.

Additional plausible causes explaining long-COVID headache include pulmonary gas-exchange abnormalities. SARS-CoV-2 is a respiratory virus that may cause bilateral pneumonia in 46.8% of patients, with a need for oxygen therapy in 69.4% or ventilatory support in 17.5% of hospitalized cases [33,34]. However, according to the ICHD-3, headache attributed to hypoxia or carbon dioxide anomalies requires a closer temporal relation. The disorders that are listed in the ICHD-3 as potential causes of this headache include high altitude, airplane travel, diving, and sleep apnea [23].

Several studies have suggested the potential role of ventilatory inefficiency as a major cause of long-COVID syndrome [35,36]. Our research group observed data suggestive of ventilation/perfusion anomalies among patients with persistent dyspnea [37]. This finding supports the hypothesis of hypocapnia or hypoxemia as a potential mechanism for long-COVID headache. However, when stratified by the presence of long-COVID headache, the pulmonary function and the cardio-pulmonary exercise test showed similar results for those with and without persistent headache. Furthermore, no desaturation with maximal effort was observed in any of the studied patients. Therefore, the persistence of long-COVID headache did not appear to be related to hypoxia or carbon dioxide abnormalities during exercise.

Finally, blood pressure parameters were lower in patients with long-COVID headache, at the baseline and after CPET. The mean increase was lower in patients with headache (peak systolic blood pressure: +17 mmHg vs. +23 mmHg; peak diastolic blood pressure: −6 mmHg vs. +5 mmHg), but values were within the normal range, both before and after exercise. For that reason, we did not find that arterial hypertension or hypotension was a potential cause for post-COVID headache [38].

### Limitations

The present study has some limitations which should be considered. Our cohort included a modest sample size of 70 COVID-19 survivors, with a 14% prevalence of long-COVID headache. This is in agreement with the existing literature but is relatively limited. The study was performed shortly after the first wave, and therefore the proportion of patients that had been hospitalized was higher than the proportion observed during the second or third waves. Routine use of the KCCQ captures how heart failure impacts patients’ lives; however, the benefit and validity of this questionnaire in this field is unexplored and has not been specifically validated. Due to the risk of contagion, patients were not evaluated during the acute phase of the disease, so cardio-pulmonary dysfunction could still partially explain headache during that phase. Finally, since this is an exploratory study, our data should be interpreted cautiously, and can only be considered as hypothesis-generating. The absence of positive results in this study should not be interpreted as evidence of the absence of differences, and future longitudinal studies are encouraged to validate our findings. Patients with a prior history of severe cardio-pulmonary dysfunction were excluded, but future studies may evaluate specifically the frequency and pathophysiology of headache in patients with a prior history of cardio-pulmonary disorders and COVID-19.

## 5. Conclusions

In this exploratory study, we did not observe that COVID-19 survivors with long-COVID headache had a higher frequency of cardio-pulmonary dysfunction, evaluated by laboratory biomarkers, resting echocardiography, cardio-pulmonary response to exercise, or pulmonary function tests.

## Figures and Tables

**Figure 1 ijerph-19-03961-f001:**
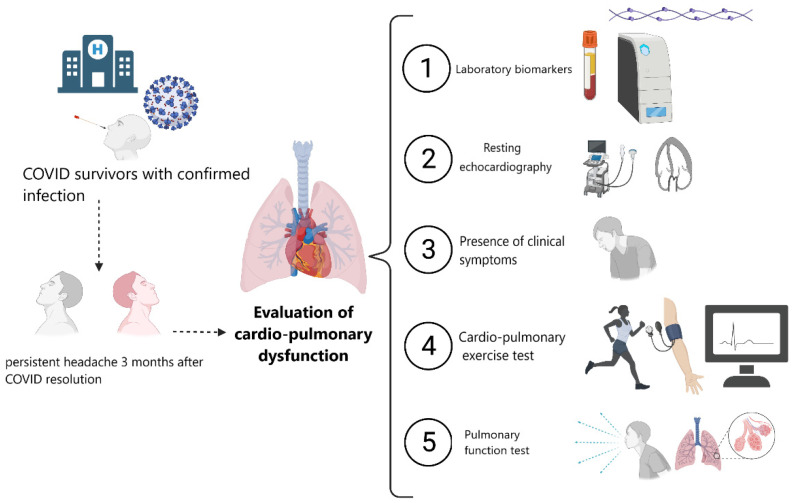
Study procedures.

**Table 1 ijerph-19-03961-t001:** Baseline characteristics of patients according to the presence or absence of long-COVID headache during follow-up.

Variable	Entire Study Sample N = 70 (100%)	Headache N = 10 (14.3%)	Non-Headache N = 60 (85.7%)	*p*-Value
Demographics				
Female sex	45 (64.3)	10 (100)	35 (58.3)	**0.011**
Age, years	54.8 ± 11.9	46.9 ± 8.45	56.13 ± 11.9	**0.023**
BMI, kg/m^2^	27.2 ± 4.6	26.43 ± 3.45	27.45 ± 4.75	0.585
BSA, m^2^	1.82 ± 0.18	1.74 ± 0.16	1.83 ± 0.18	0.118
CKD *	3 (4.4)	0	3 (5.2)	0.999
Diabetes	3 (5.9)	0	3 (5.7)	0.999
Dyslipidemia	13 (19.1)	0	13 (22.4)	0.597
Hypertension	18 (26.5)	1 (10)	17 (29.3)	0.270
Previous CAD	1 (1.5)	0	1 (1.7)	0.999
Prior pulmonary disease	5 (7.4)	0	5 (8.6)	0.999
Prior stroke/TIA	1 (1.5)	0	1 (1.7)	0.999
Long-COVID symptoms during follow-up				
Chest pain	8 (11.4)	3 (30)	5 (8.3)	0.081
Dyspnea	41 (58.6)	6 (60)	35 (58.3)	0.999
Fatigue	20 (28.6)	3 (30)	17 (28.3)	0.999
Myalgia	6 (8.6)	3 (30)	3 (5)	**0.034**
Neurological symptoms **	14 (20)	7 (70)	7 (11.7)	**<0.001**
Paresthesia	4 (5.7)	2 (20)	2 (3.3)	0.095
Olfactory abnormalities	6 (8.6)	3 (30)	3 (5.0)	**0.034**
Taste abnormalities	4 (5.7)	1 (10)	3 (5.0)	0.468
Palpitations	10 (14.3)	3 (30)	7 (11.7)	0.147

Abbreviations: BMI: body mass index; BSA: body surface area; CKD: chronic kidney disease; CAD: coronary artery disease; TIA: transient ischemic attack. * Chronic kidney disease was defined as a glomerular filtration rate of <60 mL/min or need for dialysis ** Includes paresthesia, olfactory, and taste abnormalities. Values are mean ± SD or n (%). Bold indicates significant differences (*p* < 0.05).

**Table 2 ijerph-19-03961-t002:** Follow-up complementary test of patients according to the presence or absence of long-COVID headache during follow-up.

	Entire Study Sample N = 70 (100%)	Headache N = 10 (14.3%)	Non-Headache N = 60 (85.7%)	*p*-Value
Laboratory markers				
AST (UI/L)	19 (16–25)	16 (11–22)	21 (18–20)	0.057
C-reactive protein (mg/L)	1.3 (1–2.8)	1.6 (1–2.6)	1.3 (1–2.8)	0.992
Creatinine (mg/dL)	0.84 (0.75–0.98)	0.82 (0.77–0.85)	0.85 (0.75–0.99)	0.411
D-Dimer (ng/mL)	265 (188–377)	240.5 (154–413)	267 (196–377)	0.618
Ferritin (ng/mL)	113.1 (50.1–159.1)	55.65 (30.5–123)	122 (53.3–171.3)	0.052
Fibrinogen	427 (376–491)	498 (468–504)	421 (367–466)	**0.028**
Interleukin-6 (pg/mL)	3.42 (2.6–4.4)	2.62 (2.28–3.48)	3.49 (2.63–4.45)	0.174
Haemoglobin (g/dL)	14 (13.5–15.3)	13.8 (13.6–14.7)	14.2 (13.5–15.6)	0.310
Lymphocytes (cells/mm^3^)	2185 (1800–2790)	1845 (1520–2650)	2270 (1815–2900)	0.264
NT-ProBNP (pg/mL)	41 (23–68)	50 (25–92)	39.5 (20–68)	0.669
Hs TnT (pg/mL)	5.4 (3.1–7.54)	3 (3–3.67)	5.57 (3.93–7.7)	**0.029**
Resting echocardiographic findings				
LAVI (mL/m^2^)	22.1 (17.7–27.8)	23.4 (21.2–25.4)	217 (17.6–29)	0.999
LVEF (%)	64 (59–68)	66 (61–70)	63 (58–68)	0.233
LVEDVi (mL/m^2^)	75 (66–100)	39.6 (33.1–42.4)	44.4 (38.4–54.2)	0.067
LVESVi (mL/m^2^)	16.2 (12.3–20.1)	14 (10.2–16.7)	16.5 (12.7–20.6)	0.196
Mitral E/A ratio	0.9 (0.76–1.22)	1.27 (1.06–1.48)	0.88 (0.75–1.2)	**0.010**
E/e’	6.5 (4.9–7.9)	6.27 (4.7–7.35)	6.57 (5–8.13)	0.544
TAPSE (mm)	23 (20–26)	22 (21–26)	23 (20–25)	0.853
Global longitudinal strain (%)	20 (22–19)	20 (22–19)	20 (22–19)	0.643
Kansas City Cardiomyopathy Questionnaire				
Global score	70 ± 19.42	64.53 ± 22.5	71 ± 18.8	0.335
QoL	58.4 ± 29.55	57.5 ± 33.9	58.6 ± 29	0.918
Physical function	66.9 ± 14.6	65 ± 14.4	67.28 ± 14.8	0.654
Social function	65.46 ± 26.7	49.5 ± 25.1	68.4 ± 21.2	**0.014**
Symptom score	80.7 ± 22.6	70.2 ± 21.8	82.6 ± 20.8	0.235

Abbreviations: LAVI: left atrial volume index; LVEF: left ventricular ejection fraction; LVEDVi: left ventricular end-diastolic volume index; LVESVi: left ventricular end-systolic volume index; QoL: quality of life. Values are mean ± SD or median (IQR). Bold indicates significant differences (*p* < 0.05).

**Table 3 ijerph-19-03961-t003:** Cardio-pulmonary evaluation of patients after SARS-CoV-2 infection according to the presence or absence of long-COVID headache during follow-up.

	Entire Study Sample N = 70 (100%)	Headache N = 10 (14.3%)	Non-Headache N = 60 (85.7%)	*p*-Value
Cardio-pulmonary exercise test				
Breathing reserve (%)	41 (32–51)	45 (30–50)	41 (34–52)	0.723
RER	1.11 (1.05–1.21)	1.05 (1.01–1.08)	1.12 (1.05–1.21)	0.059
Peak Vo_2_ (mL/min/kg)	19.4 (17.2–24.8)	19.9 (17.2–22.6)	19.5 (17.1–24.6)	0.754
% of predicted pVo_2_	88 (76–100)	88 (78–94)	88 (74–100)	0.820
Vo_2_ at AT_1_ (mL/min/kg)	15.4 (12–19.2)	18.7 (14.9–19.6)	15.3 (11.8–19)	0.219
% of predicted Vo_2_/HR	101 (83–110)	103 (93–108)	101 (79–110)	0.756
VE/Vco_2_ slope	30.3 (27.5–34.9)	30 (29.7–34.3)	30.6 (27–35.5)	0.942
VE/Vco_2_ at AT_1_	34.7 (32.3–39.5)	33.7 (32.5–39.5)	34.9 (32.2–39.3)	0.806
PET_CO2_ (mmHg) at AT_1_	38 (33.5–39.5)	38 (34–38)	37 (33.5–40)	0.956
% of predicted HR	90.3 (83.9–97.4)	87 (81.7–94.7)	92 (84–99)	0.385
Resting systolic BP (mmHg)	139 (124–146)	122 (109–140)	140 (126–147)	**0.020**
Peak systolic BP (mmHg)	143 (160–177)	139 (134–143)	163 (151–180)	**<0.001**
Resting diastolic BP (mmHg)	86 (77–95)	86 (75–94)	86 (75–95)	0.692
Peak diastolic BP (mmHg)	90 (81–100)	80 (77–90)	91 (85–100)	0.059
Pulmonary function				
DL_CO_ % of predicted	88.8 (80–97)	94 (85.5–97)	87 (79–95.3)	0.297
FEV1 % of predicted	112 (103.5–121.5)	114 (104–124)	112 (103–120)	0.742
FVC% of predicted	116 (105–131)	124 (114–132)	115 (104–130)	0.263
FEV1/FVC (%)	100 (91.6–105)	99 (94–103)	100 (90–105)	0.728

Abbreviations: AT: anaerobic threshold; BP: blood pressure; DL_CO_: carbon monoxide diffusion capacity; FEV1: forced expiratory volume in 1 min; FVC: forced vital capacity; HR: heart rate; VE: minute ventilation; Vo_2_: oxygen consumption. Bold indicates significant differences (*p* < 0.05).

## Data Availability

The data presented in this study are available upon reasonable request from the corresponding author. The data are not publicly available due to the inclusion of information that could compromise the privacy of research participants.

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
