# Peer review of "Cardio-Pulmonary Dysfunction Evaluation in Patients with Persistent Post-COVID-19 Headache"

_ijerph, 2022, doi:10.3390/ijerph19073961_

Round 1
Reviewer 1 Report
The manuscript was prepared very well. The introduction section justifies the purpose of the study. I congratulate the authors for the preparation of the manuscript
However, I have the following comments:
Introduction
You should include more information about long COVID.
DOI: 10.1136/bmj.n1648
The headache is caused by the sequelae of COVID? or by the symptoms of persistent COVID? clarify this issue.
You indicate that it occurs after systemic infections, which ones?
In the introduction you state that the pathophysiology of headache is unknown but then you suggest hypoxemia, hypercapnia...clarify this.
You must justify with more arguments the hypothesis, why in a period of 3 months? and not before? 3 months since the negative diagnosis?
You should improve the introduction and follow a structure that explains the long COVID, its pathophysiology, especially the headache and its relationship with the objective of the study.
Materials and Methods
The long COIVD patients, who diagnosed them, did they have other symptoms? Do they suffer from any complication or comorbidity that could be the cause of headache and not COVID? please clarify this question.
The methodology is perfectly described and carried out
Results
The tables/figures and the text describing them do not require any input, it is the strongest part of this study.
Discussion
- What specifically does this manuscript contribute?
- Include a limitations section specific.
- They should include some reference to justify their contributions, for example line 246, 258, 275.
- Paragraph 276-281, should better justify the results.
- The discussion does not address important issues to justify the results nor does it present comparisons with similar studies or with other studies describing other symptoms of long COVID. Please review the entire discussion in full.
Reviewer 2 Report
This is a study seeking association between persistent post-COVID-19 headache and disorders of circulatory and respiratory systems. The pathomechanism of this type of headache is unknown. Hence the study looking for such an association is highly relevant.
Strengths of this study include a structured collection of many parameters regarding patients cardiac and pulmonary function. Data analysis and ethical aspects seem to have been handled adequately.
The major limitation of this study is a very small study group (n=10). This disallows calculation of any but strongest associations. Consequently, any conclusions regarding lack of association are of a very low value and should not be part of a major message from this trial (as per title, conclusions).
Minor limitations include:
- P1L35-7 Headache is much more prevalent (39-72%) when analysed in prospective studies and/or by structured questionnaires:
- Lechien JR, Chiesa-Estomba CM, Place S, et al. Clinical and epidemiological characteristics of 1,420 European patients with mild-to-moderate coronavirus disease 2019. J Intern Med 2020; 288: 335–344.
- O’Keefe JB, Tong EJ, O’Keefe GD, et al. Description of symptom course in a telemedicine monitoring clinic for acute symptomatic COVID-19: a retrospective cohort study. BMJ Open 2021; 11: e044154.
- Straburzyński M, Nowaczewska M, Budrewicz S, et al (2021) COVID-19-related headache and sinonasal inflammation: A longitudinal study analysing the role of acute rhinosinusitis and ICHD-3 classification difficulties in SARS-CoV-2 infection. Cephalalgia. doi:10.1177/03331024211040753
- P1-2 L42-4 These associations are a bit more complicated: Bolay H, Karadas Ö, Oztürk B, Sonkaya R, Tasdelen B, Bulut TDS, Gülbahar Ö, Özge A, Baykan B. HMGB1, NLRP3, IL-6 and ACE2 levels are elevated in COVID-19 with headache: a window to the infection-related headache mechanism. J Headache Pain. 2021 Aug 12;22(1):94. doi: 10.1186/s10194-021-01306-7.
- P2 L47 There is no evidence associating hypercapnia etc. with COVID-19 related headache. These are just hypotheses.
- P2 L60-64 The sentence is unclear - needs reediting.
- P2 L73-4 These exclusion criteria indicate that subjects with prominent cardiac or respiratory dysfunction were excluded from the study. It is understandable why authors took these precautions, but it needs to be included in the limitation section of discussion.
- P5 L184 I was surprised by this result (9 months) and so would readers assuming from materials and methods that evaluation took place after 3 months post-COVID. The description of patients' flow should be made more comprehensible.
- Statistically siginificant differences between the groups (e.g. blood pressure, fibrinogen) could be attributed to confounders (i.e. age, sex) but are only selectivlely commented in discussion.
Reviewer 3 Report
My suggestions for authors concerne about the subtype of headache associated with sars-cov 19 infection.
Infact, in a recent paper published on Cephalalgia 2020 authors observed that migraine is the more frequently phenotype covid-19 associated.
I suggest to cyte this work
Author Response
Response to Reviewer 3 Comments, minor reviews
Point 1: My suggestions for authors concerne about the subtype of headache associated with sars-cov 19 infection. In fact, in a recent paper published on Cephalalgia 2020 authors observed that migraine is the more frequently phenotype covid-19 associated. I suggest to cyte this work
Response 1: We have included the suggested paper, thank you for your feedback.
Round 2
Reviewer 1 Report
The authors have taken all suggestions on board. For my part there is nothing more to add because the manuscript has improved in quality.
Author Response
We thank you for such positive feedback
Reviewer 2 Report
Dear Authors,
Thank you for acknowledging my comments. I applaud the overall merit of your methodology, however I still believe that 10 subjects is not enough to publish a study that found no statisticaly significant associations.
As per your response:
ad 2.
As per details the title "Persistent post-COVID headache is not associated with cardio- 2 pulmonary dysfunction" seems an extremely definite statement.
ad 3.
In no way I'd like to undermine the experience of your investigator. If you are 100% sure that each patient was asked during examination about presence of headache please state so in M&M. However, when no structured questionnaires are used it is only natural to omit less bothersome symptoms in certain situations. In blunt words: when patient is suffocating in ER, headache is the last thing that would be discussed. As per your reference choice, once again you refer to studies that collected data from retrospective chart reviews.
ad 4-5, 8
Thank you. Your response is satisfactory.
ad 6.
I'd recommend changing "ventilatory inefficiency" to 'respiratory failure' if that was the case.
ad 7.
Please bear in mind that your primary hypothesis is that cardiopulmonary dysfunction might lead to persistent headache. Meanwhile you have excluded from the study potentially large group of subjects (due to severe cardiopulmonary dysfunction). It requires a more robust discussion than the one you have provided. What were your motives to do so? What data might have been as a consequence?
